# Effect of different thresholds for CT perfusion volumetric analysis on estimated ischemic core and penumbral volumes

**Simo Karhi**[1,2]*, **Olli Tähtinen**[1,2], **Joona Aherto**[1,2], **Hanna Matikka**[1], **Hannu Manninen**[1,2], **Ossi Nerg**[3,4], **Mikko Taina**[1,2], **Pekka Jäkälä**[3,4], **Ritva Vanninen**[1,2]

**1** Department of Clinical Radiology, Kuopio University Hospital, Kuopio, Finland, **2** Unit of Radiology, Institute of Clinical Medicine, University of Eastern Finland, Kuopio, Finland, **3** Unit of Neurology, Institute of Clinical Medicine, University of Eastern Finland, Kuopio, Finland, **4** Neuro Center, Kuopio University Hospital, Kuopio, Finland

* simo.karhi@ksshp.fi

## Abstract

### Purpose

This single-center study compared three threshold settings for automated analysis of the ischemic core (IC) and penumbral volumes using computed tomographic perfusion, and their accuracy for predicting final infarct volume (FIV) in patients with anterior circulation acute ischemic stroke (AIS).

### Methods

Fifty-two consecutive AIS patients undergoing mechanical thrombectomy (November 2015–March 2018) were included. Perfusion images were retrospectively analyzed using a single CT Neuro perfusion application (syngo.via 4.1, Siemens Healthcare GmbH). Three threshold values (S1–S3) were derived from another commercial package (RAPID; iSchema View) (S1), up-to-date syngo.via default values (S2), and adapted values for syngo.via from a reference study (S3). The results were compared with FIV determined by non-contrast CT.

### Results

The median IC volume (mL) was 24.6 (interquartile range: 13.7–58.1) with S1 and 30.1 (20.1–53.1) with S2/S3. After removing the contralateral hemisphere from the analysis, the median IC volume decreased by 1.33(0–3.14) with S1 versus 9.13 (6.24–14.82) with S2/S3. The median penumbral volume (mL) was 74.52 (49.64–131.91), 77.86 (46.56–99.23), and 173.23 (125.86–200.64) for S1, S2, and S3, respectively. Limiting analysis to the affected hemisphere, the penumbral volume decreased by 1.6 (0.13–9.02), 19.29 (12.59–26.52), and 58.33 mL (45.53–74.84) for S1, S2, and S3, respectively. The correlation between IC and FIV was highest in patients with successful recanalization (n = 34, r = 0.784 for S1; r = 0.797 for S2/S3).

**Data Availability Statement:** All relevant data are within the manuscript and its Supporting Information files.

**Funding:** This study was supported by governmental funding from the Kuopio University Hospital Research Commission (grant number: 5772789). First author Simo Karhi has also received a personal study grant/stipendium from the Radiological Society of Finland / Finnish neuroradiologist association. Funders did not play any role in the study design, data collection and analysis, decision to publish, or preparation of the manuscript. There was no additional external funding received for this study.

**Competing interests:** The authors have declared that no competing interests exist.

## Conclusion

Optimizing thresholds significantly improves the accuracy of estimated IC and penumbral volumes. Current recommended values produce diversified results. International guidelines based on larger multicenter studies should be established to support the standardization of volumetric analysis in clinical decision-making.

## Introduction

The use of computed tomographic perfusion (CTP) and computed tomographic angiography (CTA) imaging for diagnosis of acute ischemic stroke has increased dramatically over the last decade [1, 2]. Several recent large-scale randomized controlled trials have further established the role of CT to help select patients for endovascular treatment (EVT) and improve their outcomes [3–9]. Information gained from CTP has also been used to support the clinical decision-making for patients admitted more than 6 h after symptom onset [10–12]. The ischemic core (IC) and hypoperfused tissue volumes obtained by CTP were shown to predict final infarct volume (FIV) determined by follow-up non-contrast CT (NCCT) or magnetic resonance imaging [13]. However, a small ischemic area seems to be the only independent predictor of good reperfusion, even after complete recanalization [14] and in some cases, the IC determined by CTP could overestimate the size of the infarct, especially in patients with early reperfusion [15, 16].

Compared with methods based on different visual scoring systems, the quantitative volumetric measurements made using automated software show better correlations with the patient outcomes [17, 18]. Individual scores estimated with the Alberta Stroke Program Early CT Score (ASPECTS), for example, may represent a wide range of absolute IC volumes due to the unequal weighting and inter-individual variability of certain brain regions that may affect patient selection [19]. In the future, fully automated and validated software for volumetric assessment could be utilized in primary treatment centers, supporting earlier decision-making and potentially reducing the time to reperfusion [20].

A comparative study showed that the accuracy of CTP analysis differs between commercial software packages when using the company's default threshold values for detecting the IC [21]. Another recent post hoc study reported that estimated IC volumes vary greatly between three postprocessing software packages [22]. Thus, the observed differences are not just dependent on the software/hardware company, because the threshold values entered into the software are usually modifiable. However, that study [22] also indicated that utilization of similar threshold levels for measurements might still not provide optimal agreement between software from different companies. In addition to differences in the ability to estimate IC volumes, the software settings recommended by each company may differ in their ability to assess the penumbral volume, another important parameter used in clinical decision-making. Furthermore, semiautomatic CTP analysis may yield results that vary with the distribution of the analyzed parenchyma, including whether the analysis was limited to the ipsilateral parenchyma or included both hemispheres.

The aims of this retrospective single-center study were 1) to evaluate the performance of a single postprocessing tool using three threshold values adapted from the default settings recommended by different software companies, 2) to compare the CTP maps obtained with these three threshold settings, and 3) to compare the CTP maps obtained using data from the ipsilateral hemisphere only vs. CTP maps created using data from bilateral hemispheres.

## Materials and methods

### Patients and endovascular procedures

The study was approved by the Research Ethics Board of the Kuopio University Hospital (KUH) (no. 5772789) and was conducted according to the principles of the Declaration of Helsinki. Patient data was anonymized prior to conduction of the study. Due to the nature of a retrospective registry study, ethical board waived the need for informed consent. Between December 2015 and March 2018, all patients admitted with AIS (Acute Ischemic Stroke, estimated n = 1400) were scheduled for head CT, CTA, and CTP. A total of 125 patients with CTA-confirmed anterior circulation large artery occlusion (LAO) and a suitable IC/perfusion profile in CTP were selected for endovascular reperfusion procedures. According to international guidelines, individual treatment decisions were based on duration of symptoms, distance to the university hospital, other underlying diseases, and the functional status prior to stroke [23, 24]. For the purpose of this study, we applied the following inclusion criteria: age ≥18 years; LAO of the anterior circulation (distal ICA; MCA M1 or M2 segment), no concomitant carotid artery stenting; and relevant imaging performed at our hospital prior to treatment. Altogether, 73 patients were excluded for the following reasons: 32 patients with no suitable imaging data for the study retrospectively available (imaging performed elsewhere, n = 23; CTP not performed, n = 5; CTP failure, n = 2; control imaging with magnetic resonance imaging, n = 2) and 27 patients for other reasons (fluctuating symptoms, n = 2; chronic intracranial stenosis in angiography, n = 5; proximal ICA stenosis requiring carotid artery stenting, n = 20). Another 14 patients were excluded due to good response to bridging therapy with intravenous thrombolysis.

For the remaining 52 patients, age, gender, and clinical background information were retrospectively collected from medical records. This included data on medically treated hypertension, diabetes, hypercholesterolemia, known atrial fibrillation, or coronary artery disease along with AIS onset, imaging and reperfusion times, and delays. Post-procedural reperfusion was retrospectively evaluated from the patient records and postprocedural angiography images. This was considered successful in patients with TICI (Thrombolysis In Cerebral Infarction) score of 2b or 3 and unsuccessful in patients with a TICI score of ≤2a [25]. The endovascular procedures are described in S1 Fig and S1 Text and S1–S3 Tables.

### Image acquisition

On admission, the patients were scanned according to a routine clinical CT stroke protocol, which included non-contrast head CT, CTP, and CTA to assess the cervicocranial arteries, followed immediately by scanning of the ascending aorta and heart. SOMATOM Definition Flash or SOMATOM Definition Edge scanners (Siemens Healthcare, Erlangen, Germany) with 64-row Stellar detectors were used with a contrast bolus of 50 mL (Omnipaque 350 mg/mL, 6 mL/s) to assess cerebral perfusion. The perfusion images were acquired every 1.5 s in the sequence shuttle mode (80 kVp, 80 mAs, 64 × 0.6 mm collimation, 0.57 s, CTDIvol = 84 mGy, total scan time 35 s), which enables brain coverage of 10 cm.

### CTP volumetric analysis

The CTP images were analyzed using CT Neuroperfusion software (syngo.via 4.1 VB30A; Siemens Healthcare GmbH, Erlangen, Germany) with automated preprocessing for motion correction, brain tissue segmentation (removal of air, bone, and cerebrospinal fluid) and vessel definition. The deconvolution method was used to compile perfusion maps with indicated parameters for cerebral blood flow (CBF), cerebral blood volume (CBV), mean transit time

(MTT), time to peak (TTP), and time to maximum ($T_{max}$) [26]. The volumes of tissues at risk (i.e., penumbra, defined as the total hypoperfused tissue − IC) and non-viable tissue (i.e., IC) were derived from these maps. The CTP analysis was first performed using bilateral data of both hemispheres, while the side with the longest time to drain was automatically characterized as the lesion side and the contralateral side was used as a reference for relative values. Then, the analyses were performed for the affected hemisphere only. A single observer (SK) entered the imaging data into the automated processing software using the following three settings:

*Setting 1 (S1)*: Threshold settings adapted from the commercial volumetric analysis package RAPID (iSchema View, Menlo Park, CA), where regions with a $T_{max}$ delay of >6 s are registered as hypoperfused tissues and the IC is derived as the volume of tissue with a relative CBF of <30% that of normal tissue [17, 21, 27–30].

*Setting 2 (S2)*: Threshold settings where regions with a CBF of <27/100 mL/min are registered as hypoperfused tissue and the IC is derived as the region with CBV <1.2/100 mL [26]. These were also the default values for deconvolution-based calculation in the CT Neuro Perfusion software.

*Setting 3 (S3)*: Thresholds settings recognizing hypoperfused tissue as a region of tissue with a CBF of <35.1/100 mL/min and the IC is defined as in S2: these were prior default setting of the software, also used in a recent reference study [21, 26, 31].

## Final infarct volume

The patients underwent a NCCT scan 1–4 days (mean 1 ± 0.75 days) after hospital admission on a SOMATOM Definition Flash scanner (Siemens Healthcare, Erlangen, Germany). The scan protocol was either a regular non–contrast helical head CT (64 × 0.6 mm collimation, 100 kVp with quality ref. mAs 322, 0.55 s rotation time, pitch 0.55, 4D dose modulation (CARE-Dose) and organ based modulation (X-care) on) or a regular head DECT (40 x 0.6 mm collimation, 80 kVp and 140 kVp with ref. mAs 360 and 180 respectively, 0.5 s rotation time, pitch 0.55 and 4D dose modulation on). The FIV was measured by an independent observer (JA) together with an experienced interventional neuroradiologist (OT), both being blinded to other clinical information. Transversal consecutive 4.5–5.0 mm thick slices of the head CT in the subacute phase were used to measure areas on a Sectra workstation (version 20.1.7.2386; Sectra, Linköping, Sweden). The FIV was calculated using Simpson's method by multiplying each manually traced infarction area by the section thickness and summing the volumes of the separate sections [32].

## Statistical analysis

Statistical analyses were performed using SPSS for Macintosh version 24 (SPSS, Chicago, IL). Continuous variables are presented as the mean and standard deviation (SD), or as the median and interquartile range (IQR) for variables showing skewed distributions. Categorical variables are presented as the absolute frequencies or percentages. When measured values did not show a normal distribution based on the Kolmogorov–Smirnov test, the Mann–Whitney *U* test was used to compare groups. Wilcoxon's signed rank test was used to detect paired differences in the IC and penumbral volumes between different threshold settings and FIV, and between differences measured from both hemispheres versus the affected hemisphere data within each setting. Additionally, differences between each setting pair were tested on one sample T-test, and pairs showing non-significant difference were evaluated using Bland Altman Plots. Pearson's $\chi^2$ test was used to determine associations between nominal variables, and Fisher's exact test was used when appropriate. P-values <0.05 were set to indicate statistically significant results.

Pearson's correlation coefficient was used to evaluate the strength of correlations with FIV in subgroups of patients. In subgroup analyses, the patients were dichotomized into two groups according to the success of the reperfusion (i.e., TICI 2b/3).

## Results

Of 125 consecutive EVT patients, 52 were included in the study. The baseline, radiological, and procedural characteristics of these patients are described in the S1 Fig and S1 Text and S1–S3 Tables.

### Ischemic Core volume

The IC volumes are listed in Table 1. When IC was measured using data from both hemispheres, the median volume was 24.6 mL (IQR 13.7–58.1 mL) with S1 and 30.1 mL (IQR 20.1–53.1 mL) with S2/S3 (paired difference $P = 0.231$). When the analysis was limited to the affected hemisphere, the median IC volumes were 20.89 mL (IQR 11.11–49.85 mL) with S1 and 19.94 mL (IQR 11.54–28.65 mL) with S2/S3 (Paired difference $P = 0.028$ in the Wilcoxon test). Differences between S1 and S2/3 volume measurements were further tested on One Sample T-test, and no difference was observed between the IC volumes when using both hemispheres for measurements ($P = 0.368$). When IC measurements were performed with the affected hemisphere only, IC volumes measured using S1 were significantly larger compared to those measured using setting S2/3 ($P = 0.006$). A similar result was observed, when performing the tests in a subgroup of patients with successful recanalization only (TICI 2b/3, n = 34) with no difference ($P = 0.279$) using both hemispheres and a significant difference ($P = 0.030$) when using the affected hemisphere only. These results are illustrated in Fig 1A–1D. A Bland Altman plot visualizes the pair S1 versus S2/3 using both hemispheres, with no significant difference between these settings on One Sample T-test (Fig 2, panel A).

### Penumbral volume

The penumbral volumes are listed in Table 1. When the penumbral volume was measured using data from both hemispheres, the median volume was 74.52 mL (IQR 49.64–131.91 mL) with S1, 77.86 mL (IQR 46.56–99.23 mL) with S2, and significantly larger at 173.23 mL (IQR 125.86–200.64) with S3. When both hemispheres were included, Wilcoxon test showed significant differences in penumbral volume between two setting pairs (S1 vs S3, and S2 vs S3,

**Table 1. Comparison of ischemic core and penumbral volumes (in mL).**

|  | Ischemic Core Volume (mL) | | | | | Penumbral volume (mL) | | | | |
|---|---|---|---|---|---|---|---|---|---|---|
|  | Both hemispheres | | Affected hemisphere | | $P^*$ | Both hemispheres | | Affected hemisphere | | $P^*$ |
|  | Median | IQR | Median | IQR |  | Median | IQR | Median | IQR |  |
| S1 | 24.6 | 13.7–58.1 | 20.89 | 11.11–49.8 | <0.001 | 74.52 | 49.64–131.91 | 70.51 | 46.59–121.57 | <0.001 |
| S2 | 30.1 | 20.1–53.1 | 19.94 | 11.54–28.65 | <0.001 | 77.86 | 46.56–99.23 | 55.97 | 31.11–75.00 | <0.001 |
| S3 | 30.1 | 20.1–53.1 | 19.94 | 11.54–28.65 | <0.001 | 173.23 | 125.86–200.64 | 110.80 | 74.69–132.54 | <0.001 |
| $P^*$, S1 vs. S2 | 0.231 |  | 0.028 |  |  | 0.316 |  | <0.01 |  |  |
| $P^*$, S1 vs. S3 | 0.231 |  | 0.028 |  |  | <0.01 |  | <0.01 |  |  |
| $P^*$, S2 vs. S3 | - |  | - |  |  | <0.01 |  | <0.01 |  |  |

$^*$Wilcoxon test

IQR indicates interquartile range; S1, setting 1; S2, setting 2; and S3, setting 3.

The CTP analysis was first performed using bilateral data of both hemispheres, then from the affected hemisphere only.

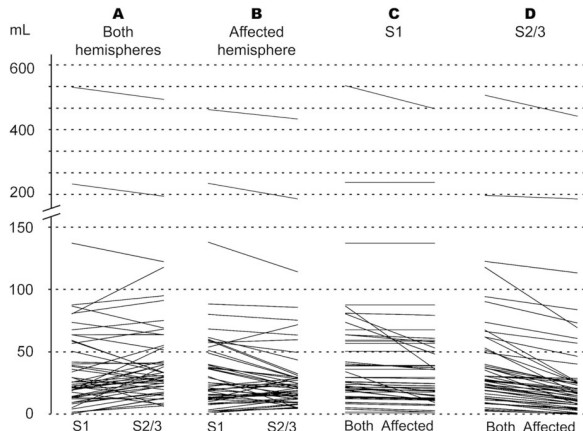

**Fig 1.** Paired comparisons of the automatically calculated ischemic core volumes (mL) for each patient measured using setting 1 (S1) versus settings 2 or 3 (S2/S3) using data from both hemispheres (A), for S1 versus S2/S3 using data from the affected hemisphere only (B); for S1 using data from the bilateral versus ipsilateral hemispheres (C), and for S2/S3 using data from the bilateral versus ipsilateral hemispheres (D).

P<0.01) and non-significant difference between S1 and S2 (P = 0.316). When the analysis was based on the affected hemisphere, the median penumbral volumes were 70.51 mL (IQR 46.59–121.57 mL) with S1, 55.97 mL (IQR 31.11–75.00) with S2, and 110.80 mL (IQR 74.69–132.54 mL) with S3. These results are illustrated in Fig 3A–3C. Using the affected hemisphere, the differences in penumbral volume between each pair (S1 vs S2, S1 vs S3, and S2 vs S3) were statistically significant using the Wilcoxon test ($P < 0.001$). Additional one Sample T-tests on the difference between the penumbral volumes measured with each setting were performed, and in line with the Wilcoxon test, no significant difference between S1 and S2 using both hemispheres was observed ($P = 0.093$)—this setting pair is further visualized using Bland Altman plot (Fig 2, Panel B). Difference between all other pairs showed statistically significant differences ($P < 0.01$).

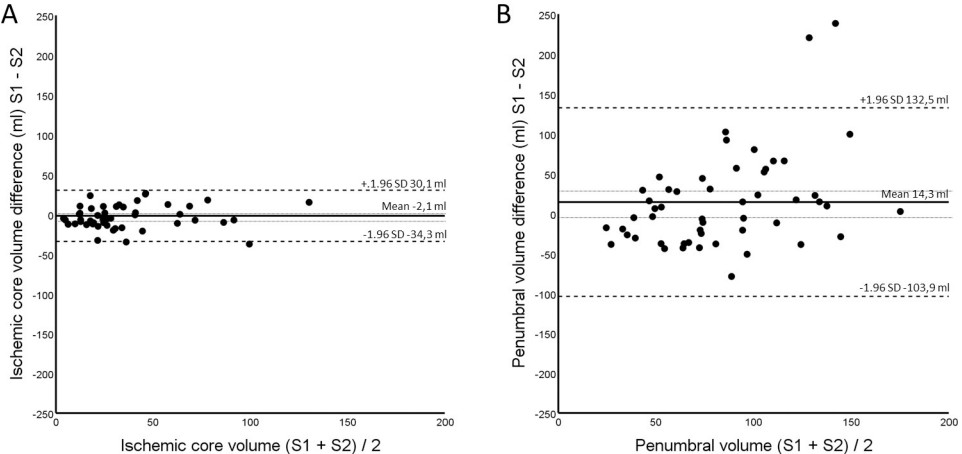

**Fig 2. Bland Altman plot visualizing the relationship of the differences of measured volumes between those setting pairs with no significant differences detected using One Sample T tests.** A: Measured ischemic core differences between setting 1 (S1) versus setting 2 (S2) using data from both hemispheres. B: Measured penumbral differences using S1 versus S2 with data from both hemispheres. Upper and lower 95% confidence limits for the mean difference in volumes are visualized using thinnest dotted lines.

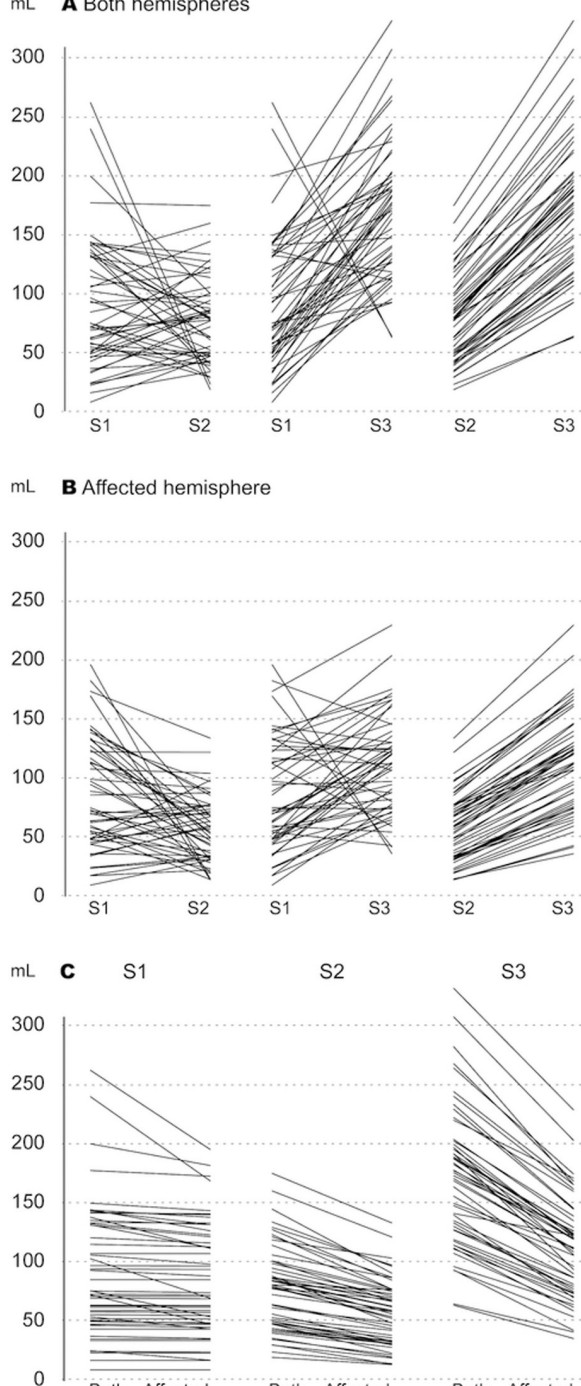

**Fig 3.** Paired comparisons of the automatically calculated penumbral volumes (mL) for each patient for setting 1 (S1) versus setting 2 (S2), S1 versus setting 3 (S3), and S2 versus S3 using data from both hemispheres (A); S1 versus S2, S1 versus S3, and S2 versus S3 using data from the affected hemisphere only (B); and S1, S2, and S3 using data from the bilateral versus ipsilateral affected hemispheres (C).

### Bilateral vs. affected hemisphere

In paired comparisons using data from both hemispheres versus data from the affected hemisphere, the IC volumes differed significantly within S1 and S2/S3 ($P < 0.001$), and the

estimated IC volumes were larger when both hemispheres were included. Compared to bilateral measurement, using only the affected hemisphere resulted in a median volume reduction of 1.33 mL (IQR 0–3.14) for S1. For S2/3, the volume difference was 9.13 mL (IQR 6.24–14.82 mL). The penumbral volume was smaller when calculated using the affected hemisphere compared with bilateral hemispheres, with significant differences for each setting ($P < 0.001$). When comparing the different settings, the volume loss was 1.6 mL (IQR 0.13–9.02 mL) with S1 and 19.29 mL (IQR 12.59–26.52 mL) with S2 and was greatest with S3 at 58.33 mL (IQR 45.53–74.84 mL). The volume distributions can be visualized in Figs 3 and 4.

## Association between estimated IC volume and final infarct volume

The association between IC and FIV was analyzed using the patient group with a successful recanalization (TICI 2b/3, n = 34). Correlation coefficients were also tested on the whole patient population for comparison. Associations between the ischemic core volume and final infarct volume can be found in Table 2.

Including both hemispheres, the median difference between the FIV and IC volume (FIV − IC) was -3.9 mL (IQR −16.30 to 20.55) with S1. In 15 patients with overestimation, the mean overestimation compared to FIV was 64% with a median excess volume of 15 mL (IQR 7.05–27.67 mL). When limited to the affected hemisphere, the median difference was -0.89 mL (IQR −15.24 to 23.98 mL), and in 18 cases with overestimation the mean overestimation in FIV was 63% and the median excess volume was 14.63 mL (IQR 5.84–18.65 mL). Using S2 or S3, the median difference between IC and FIV using data from both hemispheres was -13.53 mL (IQR −26.93 to 27.41). The mean overestimation was 72% in 21 patients with overestimation and the median excess volume was 23.36 mL (15.23–32.86 mL). When measuring the affected side only with these settings, the median difference between the FIV and IC volume was -1.65 mL (IQR −12.48 to 37.38 mL). In 18 patients with overestimation, the average overestimation in FIV was 65% with a median overestimation volume of 11.00 mL (IQR 6.96–19.16 mL).

The correlation between IC volume and FIV was strong (r = 0.774 using both hemispheres and r = 0.784 using the affected hemisphere only) using Setting S1, very similar to S2/S3 (r = 0.751 using both hemispheres and r = 0.797 using the affected hemisphere only). Scatter plot figures visualizing the correlation between IC and FIV can be found in S1 Fig and S1 Text and S1–S3 Tables.

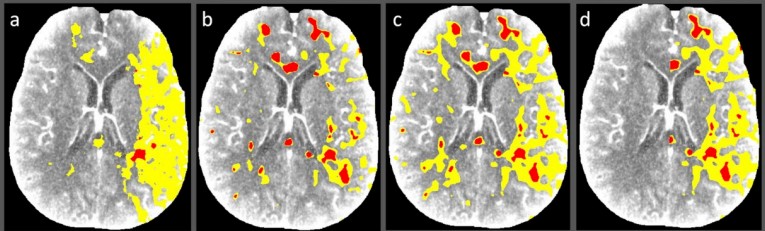

**Fig 4. Illustrative computed tomographic perfusion map images for a patient with ischemic stroke.** (A–C) Data were postprocessed using the threshold values adapted to Syngo.via from another commercial software package (setting 1) (A), up-to-date company-recommended threshold values for Syngo.via (setting 2) (B), and older Syngo.via default parameters used in a reference study (setting 3) (C). (D) Limiting the analysis to the affected hemisphere using setting 3 yields a sensible, but faulty estimate of the penumbra and ischemic core volumes, potentially resulting in inaccurate diagnosis.

**Table 2. Associations between the ischemic core volume and final infarct volume in all patients and those patients with a successful recanalization (TICI 2b/3).**

| | All patients | | | | Patients with TICI 2b/3 | | | |
|---|---|---|---|---|---|---|---|---|
| | IC (mL) | FIV−IC (mL) | r | P* | IC (mL) | FIV−IC (mL) | r | P* |
| S1 both hemispheres | 24.6 (13.7–58.1) | 0.16 (−14.8 to 19.3) | 0.663 | 0.636 | 24.0 (13.9–63.7) | −3.9 (−16.3 to 20.55) | 0.774 | 0.952 |
| S1 affected hemisphere | 20.9 (11.1–49.8) | 2.96 (−13.9 to 23.6) | 0.693 | 0.161 | 20.9 (12.2–58.0) | −0.89 (−15.2 to 24.0) | 0.784 | 0.499 |
| S2/3 both hemispheres | 30.1 (20.1–53.1) | −8.1 (−26.2 to 23.7) | 0.639 | 0.906 | 33.2 (21.4–63.8) | −13.5 (−26.9 to 27.41) | 0.751 | 0.993 |
| S2/3 affected hemisphere | 19.9 (11.5–28.7) | 3.06 (−8.6 to 33.6) | 0.670 | 0.049 | 22.7 (12.0–44.2) | −1.6 (−12.5 to 37.4) | 0.797 | 0.169 |

Values are presented as median (interquartile range).

*Wilcoxon test between Ischemic Core Volume vs Final Infarct Volume. IC indicates ischemic core; FIV, final infarct volume; S1, setting 1; S2, setting 2; S3, setting 3; and *r*, Pearson's correlation coefficient TICI = Thrombolysis in Cerebral Infarction Scale.

Wilcoxon test comparing both hemispheres vs affected hemisphere only showed significant differences between each of the pairs within each setting (P<0.001).

## Discussion

Significant differences have been reported for the abilities of different commercial perfusion CT software packages to predict the final infarct volume after EVT when using the company-recommended default threshold values [21, 22]. The present study used a fully automated software package with three different threshold settings to quantify CTP data. S1, which used parameters obtained from commercial volumetric software used in several recent studies [5, 7, 11, 12, 13], was tested against company-specified up-to-date values (i.e. S2) and prior default threshold values (i.e. S3) [26]. We compared the performances of these three settings in a population of patients with unilateral AIS who had undergone mechanical thrombectomy with modern recanalization techniques according to international guidelines. We found remarkable differences especially in the estimated penumbral volumes between the three tested threshold values. When assessing the potentially rescuable penumbral volume, we found a 2-fold difference between the widely used setting (i.e. S1) [17, 27–30] and the adapted setting (i.e. S3) [21, 26].

Another key finding was the observation of significant differences in results when using data from both hemispheres versus data from the affected hemisphere only. However, the median volume change was only 1.6 mL with S1 versus 58.33 mL with S3 when measuring penumbral tissue–demonstrating that a large proportion of the detected penumbral tissue could be, in fact, an artefact reflecting from the use of the healthy hemisphere when using the latter setting. In general, the threshold levels included in S3 showed inferior results, most likely because the values are based on earlier studies using the maximum slope method rather than the contemporary deconvolution method [33, 34]. These values were recently updated to better correspond to the results obtained using the deconvolution method, to provide a more accurate estimation of penumbral volume [26].

Despite the equally high correlations between the estimated IC volumes determined using each threshold setting and the follow-up FIV, the spatial distributions of the tissue volumes differed visually when the analysis was limited to the affected hemisphere, indicating a variable amount of error in the detected Ischemic tissue between the thresholds (similar to the one visualized when comparing the penumbral volumes). The use of perfusion data from the affected hemisphere only resulted in a significant improvement in the accuracy of the estimated IC when compared to the FIV seen on control imaging. However, including the unaffected hemisphere may increase the reliability of the analysis because this helps the operator visualize the signal noise and can be used to generate a 'negative control' for each patient to assist interpretation.

S1 estimated the IC based on a relative CBF of <30% of the CBF in healthy tissue in the same patient–an approach widely addressed as superior to the CBV-based detection of IC [27,

35, 36]. By contrast, S2 and S3 also rely on an absolute threshold level of perfusion, expressed as a CBV of <1.2/100 mL/min. This threshold was originally based on a relative CBV value (<40% of the contralateral hemisphere) for detecting the IC [26]. Despite cerebral autoregulation, cerebral blood flow may vary between individuals even under normal conditions and is affected by pathologies such as hypertension or the degree of cervicocranial artery stenosis [37, 38]. Therefore, measuring the relative perfusion level of the healthy brain parenchyma in the same patient has some advantages compared with relying on the absolute cerebral blood volume in individuals without concomitant diseases. This principle may be adapted to a real-life clinical environment of patients suffering from multiple diseases. Future multicenter studies should describe how the volume of the healthy brain parenchyma is defined by the software used.

However, without international guidelines addressing the relative threshold levels, clinical decision-making should be undertaken with caution owing to the possibility of false-positive CTP results. In the present study, measurements that included the unaffected and affected hemispheres significantly increased the estimated volume of hypoperfused tissue compared with measurements using the affected hemisphere alone. In addition, the IC volumes were overestimated regardless of the setting used, as in prior studies [15, 16]. As expected, the correlation between the estimated IC volumes on CTP and FIV was highest in the subgroup of patients with successful recanalization [3–7, 11, 12].

All imaging was performed consistently using the same equipment and settings, and postprocessing of the imaging data obtained on admission was done using a single software package. This study design was essential to allow us to focus on the effects of different threshold values when assessing the CTP values in the study population. Potential limitations include the current equipment, which limits brain coverage to 10 cm during CTP imaging and could lead to underestimation of the loss of IC or penumbral volume. Recent studies have also shown that the optimal threshold values could vary depending on the time from the symptom onset [39, 40, 41] which could be used to further improve the accuracy of the results presented. In our study population, DWI-MRI imaging was not available, and the volume of FIV could have been affected by the varying amount of subsequent edema and swelling of he involved tissue between time scale of 1–4 days, during which the follow-up imaging was performed. In addition, the study included a small patient population due to the selection criteria, which involved unilateral EVT-treated AIS patients. Therefore, further studies in larger patient populations are encouraged.

## Conclusions

Our findings support the hypothesis that the postprocessing parameters (threshold values that quantify perfusion and hemisphere settings) may significantly affect the IC, hypoperfused tissue, and penumbral volumes determined by CTP. The induced variation was similar to or even greater than the differences observed between commercial software packages. In addition, the estimated volumes were significantly affected by including either bilateral hemispheres or the ipsilateral cerebral hemisphere in the analysis. Further studies and comparisons of commercial software packages are encouraged to develop identical and optimized parameters for CTP. Standardization of CTP parameters is also essential to establish international consensus guidelines to support clinical decision-making in the management of stroke patients.

## Supporting information

**S1 Fig. Scatter plot chart comparing ischemic core to final infarct volume (mL) in those patients that underwent successful recanalization (n = 34).** S1 = Setting 1 (panel A and B, using both hemispheres and affected hemisphere only and S2/3 = Setting 2 and 3 (Panel C and

D, using both hemispheres and affected hemisphere only).
(TIF)

**S1 Text. Depiction of endovascular procedures.**
(PDF)

**S1 Table. Excluded patients.**
(PDF)

**S2 Table. Patient baseline and admission imaging characteristics.**
(PDF)

**S3 Table. Procedural and outcome characteristics.**
(PDF)

## Acknowledgments

We would like to thank Tuomas Selander for statistical guidance.

## Author Contributions

**Conceptualization:** Simo Karhi, Olli Tähtinen, Hanna Matikka, Mikko Taina, Pekka Jäkälä, Ritva Vanninen.

**Data curation:** Simo Karhi, Olli Tähtinen, Joona Aherto, Hanna Matikka, Ossi Nerg, Mikko Taina, Ritva Vanninen.

**Formal analysis:** Simo Karhi, Olli Tähtinen, Hanna Matikka, Mikko Taina.

**Funding acquisition:** Hannu Manninen, Pekka Jäkälä, Ritva Vanninen.

**Investigation:** Simo Karhi, Olli Tähtinen, Joona Aherto, Hanna Matikka, Ossi Nerg, Mikko Taina, Ritva Vanninen.

**Methodology:** Simo Karhi, Olli Tähtinen, Hanna Matikka, Mikko Taina, Ritva Vanninen.

**Project administration:** Olli Tähtinen, Hanna Matikka, Hannu Manninen, Mikko Taina, Pekka Jäkälä, Ritva Vanninen.

**Resources:** Hannu Manninen, Ossi Nerg, Pekka Jäkälä, Ritva Vanninen.

**Software:** Simo Karhi, Olli Tähtinen, Hanna Matikka, Mikko Taina, Ritva Vanninen.

**Supervision:** Hanna Matikka, Hannu Manninen, Mikko Taina, Pekka Jäkälä, Ritva Vanninen.

**Validation:** Hannu Manninen, Ritva Vanninen.

**Visualization:** Simo Karhi, Olli Tähtinen, Hanna Matikka, Ossi Nerg, Mikko Taina.

**Writing – original draft:** Simo Karhi, Olli Tähtinen, Hanna Matikka, Ritva Vanninen.

**Writing – review & editing:** Simo Karhi, Olli Tähtinen, Hanna Matikka, Mikko Taina, Ritva Vanninen.

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
