## [Decision Letter · Decision Letter 0]

2 Nov 2020

PONE-D-20-23892

Effect of different thresholds for CT perfusion volumetric analysis on estimated ischemic core and penumbral volumes

PLOS ONE

Dear Dr. Karhi,

Thank you for submitting your manuscript to PLOS ONE. After careful consideration, we feel that it has merit but does not fully meet PLOS ONE’s publication criteria as it currently stands. Therefore, we invite you to submit a revised version of the manuscript that addresses the points raised during the review process.

We look forward to receiving your revised manuscript.

Kind regards,

Ona Wu

Academic Editor

PLOS ONE

Journal Requirements:

2. In ethics statement in the manuscript and in the online submission form, please provide additional information about the patient records used in your retrospective study. Specifically, please ensure that you have discussed whether all data were fully anonymized before you accessed them and/or whether the IRB or ethics committee waived the requirement for informed consent. If patients provided informed written consent to have data from their medical records used in research, please include this information.

"This study was supported by governmental funding from the Kuopio University Hospital Research Commission (grant number: 5772789). Funders did not play any role in the study design, data collection and analysis, decision to publish, or preparation of the manuscript.".

i) Please provide an amended statement that declares *all* the funding or sources of support (whether external or internal to your organization) received during this study, as detailed online in our guide for authors at http://journals.plos.org/plosone/s/submit-now.  Please also include the statement “There was no additional external funding received for this study.” in your updated Funding Statement.

ii) Please include your amended Funding Statement within your cover letter. We will change the online submission form on your behalf.

Reviewers' comments:

Reviewer's Responses to Questions

**Comments to the Author**

1. Is the manuscript technically sound, and do the data support the conclusions?

Reviewer #1: Partly

Reviewer #2: No

2. Has the statistical analysis been performed appropriately and rigorously? 

Reviewer #1: I Don't Know

Reviewer #2: No

3. Have the authors made all data underlying the findings in their manuscript fully available?

Reviewer #1: Yes

Reviewer #2: No

4. Is the manuscript presented in an intelligible fashion and written in standard English?

Reviewer #1: Yes

Reviewer #2: Yes

5. Review Comments to the Author

Reviewer #1: In this retrospective study, the authors assessed the differences in the CTP-derived infarction core and ischecmic penumbra volumes using 3 different threshold combinations (1- CBF <30% and Tmax> 6s; 2- CBV<1.2/100 mL and CBF< 27/100 mL/min; and 3- CBV<1.2/100 mL and CBF< 35/100 mL/min) in 52 consecutive patients with acute ischemic stroke and anterior circulation large vessel occlusion who underwent endovascular thrombectomy. They also compared the CTP-derived infarction core volumes with the final infarction size measured on the follow up non-contrast CT performed 1-4 days after admission. 34/52 patients had successful recanalization (TICI-2b or 3).

The manuscript is well-written but I have a few issues with the study aims and methods:

1- The main aim of this study was to assessed the accuracy of the CTP-derived infarction core and ischemic penumbra, which has been extensively studied and reported in the literature (for example see the below references). As expected the authors found large differences in the estimated penumbral volume. I believe the findings of this study neither enhance the current knowledge about the inaccuracy of CTP-derived infarction core and penumbra volumes nor provide a tool to overcome the CTP inaccuracies.

- Kudo K, et al. Differences in CT perfusion maps generated by different commercial software: quantitative analysis by using identical source data of acute stroke patients. Radiology. 2010;254(1):200-209.

- Kamalian S, et al. CT cerebral blood flow maps optimally correlate with admission diffusion-weighted imaging in acute stroke but thresholds vary by postprocessing platform. Stroke. 2011;42(7):1923-1928.

- Kamalian S, et al. CT perfusion mean transit time maps optimally distinguish benign oligemia from true "at-risk" ischemic penumbra, but thresholds vary by postprocessing technique. AJNR Am J Neuroradiol. 2012;33(3):545-549.

- Fahmi F, Marquering HA, Streekstra GJ, et al. Differences in CT perfusion summary maps for patients with acute ischemic stroke generated by 2 software packages. AJNR Am J Neuroradiol. 2012;33(11):2074-2080.

- Kudo K, Christensen S, Sasaki M, et al. Accuracy and reliability assessment of CT and MR perfusion analysis software using a digital phantom. Radiology. 2013;267(1):201-211. doi:10.1148/radiol.12112618

- Copen WA, et al. In patients with suspected acute stroke, CT perfusion-based cerebral blood flow maps cannot substitute for DWI in measuring the ischemic core. PLoS One. 2017;12(11):e0188891.

Further more the CTP thresholds are likely time-dependent:

- Bivard A, Kleinig T, Miteff F, et al. Ischemic core thresholds change with time to reperfusion: A case control study. Ann Neurol. 2017;82(6):995-1003. doi:10.1002/ana.25109

- Qiu W, et al. Confirmatory Study of Time-Dependent Computed Tomographic Perfusion Thresholds for Use in Acute Ischemic Stroke. Stroke. 2019;50(11):3269-3273.

- Yoshie T, et al. Perfusion Parameter Thresholds That Discriminate Ischemic Core Vary with Time from Onset in Acute Ischemic Stroke [published online ahead of print, 2020 Aug 27]. AJNR Am J Neuroradiol. 2020;10.3174/ajnr.A6744.

2- The authors used CBV for infarction core volume estimation in 2 of their 3 threshold combinations. Although they found a relatively high correlation between the estimated infarction core volumes and the follow-up final infarction size, they reported that the spatial distributions of the tissue volumes differed visually (especially when excluding the contralaterl hemisphere for analysis). There are a few issues with their methods in regard to infarction core estimation:

a- More recent studies showed thresholded-CBF is perhaps a better method for infarction core estimation. I suggest updating the results by adding more threshold combinations with inclusion of CBF-derived core volumes based on the recommendations for software used in this study (CT Neuroperfusion software).

- Kamalian S, et al. CT cerebral blood flow maps optimally correlate with admission diffusion-weighted imaging in acute stroke but thresholds vary by postprocessing platform. Stroke. 2011;42(7):1923-1928.

- Campbell BC, Christensen S, Levi CR, et al. Cerebral blood flow is the optimal CT perfusion parameter for assessing infarct core. Stroke. 2011;42(12):3435-3440.

- Bivard A, Spratt N, Levi C, Parsons M. Perfusion computer tomography: imaging and clinical validation in acute ischaemic stroke. Brain. 2011;134(Pt 11):3408-3416. doi:10.1093/brain/awr257

b- I am not sure if a Pearson correlation test is appropriate for analysis of the infarction core volumes in this study in the absence of a figure showing the scatterplots to support the results. A critical presumption for correlation/linear regression models is homoscedasticity. A correlation test can produce an erroneously high correlation coefficient when a few outliers are present, meanwhile the other data-points (usually smaller values near the center of the scatter plot) are poorly correlated. I recommend addition of the correlation scatterplots and consultation with a statistician.

c- Ideally the CTP-derived infarction core volumes should be compared to a closely acquired DWI-MRI as the gold standard test. Because MRI is not always available or not usually performed initially to avoid delay in endovascular treatment, many studies use the final infarct size after endovascular treatment with presumption of minimal or no significant infarct growth following successful recanalization. Therefore, I suggest to redo the correlation study between the CTP-derived infarction core volume and final infarct size only in the 34 patients who achieved successful recanalization (TICI-2b or 3).

Reviewer #2: In this study the authors point out the lack of standardization for measuring ischemic core and penumbra volumes using CT perfusion thresholds and software analysis. Their aim is to show the accuracy for predicting final infarct volume on a CT between one and four days following stroke for each of the CT perfusion outputs which include threshold values that are used by RAPID Ischemia View and in house proprietary methods. They show that with each CTP software output that there is a high degree of variance with respect to the comparison of infarct core penumbra on final infarct volume comparisons for each of these software analyses. The manuscript is of importance but has several major limitations which the authors should address.

1. The authors excluded 73 patients from their analysis. Some of their criteria appear to be less than explanatory for instance 32 patients with “deviant imaging data” were excluded. Do the authors mean that the images are of poor quality and could not be analyzed. Why were patients with fluctuating symptoms excluded as well as those with chronic inter-cranial stenosis. The authors should refine their inclusion and exclusion criteria.

2. It is common for CT type analyses to compare their accuracy of ischemic core and penumbra thresholds with a final infarct volume usually DWI at 24 hours but with respect to this manuscript the issue is not so much how the various techniques relate to final infarct volume but how they compare between each other. Therefore, I would recommend that the authors compare the infarct core volume and penumbra volumes between each imaging software technique. This analysis can be performed by Bland Altman plots.

3. It is not clear whether the authors controlled for recanalization status because the final infarct volume for subjects with early recanalization will be potentially smaller than if they had not recanalized. In order to obtain a more standardized comparison between the software techniques it might be most appropriate to refine the analysis to those patients that have successful recanalization with EVT.

4. The comparison with reported Rapid thresholds, is of course, not the same as comparing directly with CTP outputs provided by Rapid. Can the others explains why this was not done?

6. PLOS authors have the option to publish the peer review history of their article (what does this mean?). If published, this will include your full peer review and any attached files.

Reviewer #1: No

Reviewer #2: No

---

## [Author Response · Author response to Decision Letter 0]

16 Dec 2020

For a full response letter to the reviewers including the comments and responses, please see the individually uploaded file: "Response to the reviewers".

Reviewer #1:

Comment: "The main aim of this study was to assessed the accuracy of the CTP-derived infarction core and ischemic penumbra, which has been extensively studied and reported in the literature (for example see the below references). As expected the authors found large differences in the estimated penumbral volume. I believe the findings of this study neither enhance the current knowledge about the inaccuracy of CTP-derived infarction core and penumbra volumes nor provide a tool to overcome the CTP inaccuracies."

Response: Although not implemented in our study, we agree that the time dependency of the threshold values is a notable factor, as pointed out by the reviewer. This was added in the discussion chapter focusing on the limitations of the study, including these given references. The goal of the present study is, indeed, to underline the current inaccuracies and differing results produced by the automated analysis using the suggested thresholds given by two large software manufacturers. Although the DWI-based approach to detect ischemic core is currently preferred over CT perfusion, the latter is still widely used in the detection of acute ischaemia due to the speed and practicality. The study presented pursuits to create an objective review on the current situation of the still evolving method of automatically detecting Ischemic Core and Penumbra from diagnostic imaging using CT modality.

Comment: "2- The authors used CBV for infarction core volume estimation in 2 of their 3 threshold combinations. Although they found a relatively high correlation between the estimated infarction core volumes and the follow-up final infarction size, they reported that the spatial distributions of the tissue volumes differed visually (especially when excluding the contralateral hemisphere for analysis). There are a few issues with their methods in regard to infarction core estimation:

a- More recent studies showed thresholded-CBF is perhaps a better method for infarction core estimation. I suggest updating the results by adding more threshold combinations with inclusion of CBF-derived core volumes based on the recommendations for software used in this study (CT Neuroperfusion software)"

Response: In line with the reviewers comment, CBF-based approach of the RAPID-adapted threshold values (Ischemic core is diagnosed if the relative CBF is <30% of that in normal tissue.) showed the best visual/spatial correlation between IC, penumbral area and FIV. The aforementioned references are now added to the Discussion -section of the article. Although the settings within the software used in the study can be altered to match the aforementioned rival software, according to our current knowledge CT Neuroperfusion software for Syngo Via has no other suggested default threshold values for a similar CBFbased approach for detection of IC.

Comment: "b- I am not sure if a Pearson correlation test is appropriate for analysis of the infarction core volumes in this study in the absence of a figure showing the scatterplots to support the results. A critical presumption for correlation/linear regression models is homoscedasticity. A correlation test can produce an erroneously high correlation coefficient when a few outliers are present, meanwhile the other data-points (usually smaller values near the center of the scatter plot) are poorly correlated. I recommend addition of the correlation scatterplots and consultation with a statistician"

Response: Thank you for this comment. Scatterplots have now been created from the subgroup of TICI 2b/3 patients (n=34) with both IC thresholds.The plots can be found in the supplementary materials. A statistician was also consulted with the freshly created scatter plots considering the Pearson’s correlation test. A nonparametrical Spearman’s correlation test was performed with a similar P-value of <0.001 as a result. As we found the data being linear, our statistician supported the use of Pearson’s correlation coefficient in the analysis.

Comment: "c- Ideally the CTP-derived infarction core volumes should be compared to a closely acquired DWI-MRI as the gold standard test. Because MRI is not always available or not usually performed initially to avoid delay in endovascular treatment, many studies use the final infarct size after endovascular treatment with presumption of minimal or no significant infarct growth following successful recanalization. Therefore, I suggest to redo the correlation study between the CTP-derived infarction core volume and final infarct size only in the 34 patients who achieved successful recanalization (TICI-2b or 3):"

Response: The chapter describing the correlation between Ischemic Core and Follow-up infarct size has now been rewritten using only the patient group with a successful recanalization. The table visualizing correlation between Ischemic Core and follow-up findings has now been streamlined and is focused only on the subgroup that achieved successful recanalization (TICI 2b or 3, n=34), leaving out the other additional subgroups, as suggested. However we left the ”All patients” group in the table for comparison, as it seemed to emphazise the effect of successfulness of the treatment. 

Reviewer #2:

Comment: "1. The authors excluded 73 patients from their analysis. Some of their criteria appear to be less than explanatory for instance 32 patients with “deviant imaging data” were excluded. Do the authors mean that the images are of poor quality and could not be analyzed. Why were patients with fluctuating symptoms excluded as well as those with chronic inter-cranial stenosis. The authors should refine their inclusion and exclusion criteria."

Response: Thank you for the important comment. The term ”deviant imaging data” on page 4 has now been switched to a phrase: ”no suitable imaging data for the study retrospectively available”. The rest of the explanatory information contained in the same phrase was left unchanged: “(imaging performed elsewhere, n = 23; CTP not performed, n = 5; CTP failure, n = 2; control imaging with magnetic resonance imaging, n = 2”), in the hope of answering the issue presented by the reviewer. The similar explanatory exclusion chart can also be found on our supplementary material provided with the manuscript. It is true that the exclusion of the patients with a chronic intracranial or carotid stenosis or fluctuation of the symptoms takes the study design further away from the optimal “real world” situation. However, we wanted to exclude the patients suffering from “acute-onchronic” type of stroke etiology. In the presence of chronic intracranial artery or internal carotid artery stenosis , cerebral blood flow can be altered either by enhanced collateral circulation or, in contrast, circulation to the ischaemic area can be impaired already prior to the stroke. As our study focused solely on cerebral blood flow and cerebral blood volume on acute anterior circulation large artery occlusion, our goal was to minimize the possible confusing effects of these pathologies to the CT Perfusion imaging results. The same principle was adopted while excluding the patients who had fluctuation of the symptoms.

Liebeskind DS. Collateral circulation. Stroke. 2003 Sep;34(9):2279-84. doi: 10.1161/01.STR.0000086465.41263.06. Epub 2003 Jul 24. PMID: 12881609. https://pubmed.ncbi.nlm.nih.gov/12881609/

Van der Heyden J, Waaijer A, Van Wouter ES, van Neerven D, Sonker U, Suttorp MJ, Bal ET, Prokop M. CT measurement of changes in cerebral perfusion in patients with asymptomatic carotid artery stenosis undergoing carotid stenting prior to cardiac surgery: "proof of principle".

EuroIntervention. 2011 Apr;6(9):1091-7. doi: 10.4244/EIJV6I9A190. PMID: 21518682. Trojanowska A, Drop A, Jargiello T, Wojczal J, Szczerbo-Trojanowska M. Changes in cerebral hemodynamics after carotid stenting: evaluation with CT perfusion studies. J Neuroradiol. 2006 Jun;33(3):169-74. doi: 10.1016/s0150-9861(06)77255-8. PMID: 16840959.

Mosqueira AJ, Pumar JM, Arias S, Rodríguez-Yáñez M, Blanco Ulla M, Vázquez Herrero F, Castillo J. False ischaemic penumbras in CT perfusion in patients with carotid artery stenosis and changes following angioplasty and stenting. Neurologia. 2020 Jan-Feb;35(1):24-31. English, Spanish. doi: 10.1016/j.nrl.2017.06.002. Epub 2017 Sep 1. PMID: 28865944.

Comment: "2. It is common for CT type analyses to compare their accuracy of ischemic core and penumbra thresholds with a final infarct volume usually DWI at 24 hours but with respect to this manuscript the issue is not so much how the various techniques relate to final infarct volume but how they compare between each other. Therefore, I would recommend that the authors compare the infarct core volume and penumbra volumes between each imaging software technique. This analysis can be performed by Bland Altman plots."

Response: Thank you for this valuable comment. Results have now been expanded accordingly, and eligible Bland Altman Plots have now been created based on One Sample T-tests, which found non-significant difference between penumbral volumes measured using S1 and S2 with both hemispheres included (P=0.093) and Ischemic Core volumes measured using S1 and S2/S3 with both hemispheres included (P=0.368). Additionally, we included a Bland Altman plot using only those patients with successful recanalization (Ischemic Core between S1 and S2/S3, P=0.279) as suggested further in comment 3. Rest of the pairs showed significance (P<0.05) when their difference was tested on One Sample T-test, and therefore were not included in the Bland Altman Plot analysis. However, the previously created ladder charts visualizing the paired volume differences are left unchanged to demonstrate also the other significantly different threshold pairs. 

Comment: "3. It is not clear whether the authors controlled for recanalization status because the final infarct volume for subjects with early recanalization will be potentially smaller than if they had not recanalized. In order to obtain a more standardized comparison between the software techniques it might be most appropriate to refine the analysis to those patients that have successful recanalization with EVT."

Response: We have now expanded the phrase in page 5 describing the study methods considering retrospective evaluation of the postprocedural reperfusion: ”Post-procedural reperfusion was retrospectively evaluated from the patient records and angiographic x-ray imaging. This was considered successful in patients with TICI (Thrombolysis in Cerebral Infarction Scale) score of 2b or 3 and unsuccessful in patients with a TICI score of ≤2a.” According to the Reviewer’s suggestion, Bland Altman plots of Ischaemic Core volumes have now been created including only the successfully recanalized patient group. Penumbral volume comparison was left unchanged when comparing techniques between each other. In the chapter ”Association Between Estimated IC Volume and Final Infarct Volume” (page 11) we have now focused only on the group that achieved successful recanalization (= TICI 2b/3, n=34), as suggested. Correlation coefficients between Ischemic Core and Final Infarct Volume are calculated with the patient group that achieved successful recanalization (n=34), and scatterplot figures using only successfully recanalized patients have now also been created and can be found in the supplementary material files.

Comment: "4. The comparison with reported Rapid thresholds, is of course, not the same as comparing directly with CTP outputs provided by Rapid. Can the others explains why this was not done?"

Response: We agree on the benefits of using the original RAPID Software for comparison. Unfortunately, the RAPID commercial package is not available at our tertiary hospital. Some comparisons between vendor softwares have been published. Instead, our study aimed to analyze whether it is possible to obtain equivalent results on a single software platform by altering the CTP thresholds. This kind of comparison is clinically important, because different hospitals use different scanners from different vendors for stroke imaging, and showing that different software platforms could run similar measurements would be optimal and could help adopting standardized thresholds for hypoperfused and non-viable tissue, despite different scanners, acquisition techniques, and post-processing software settings.

---

## [Decision Letter · Decision Letter 1]

9 Feb 2021

PONE-D-20-23892R1

Effect of different thresholds for CT perfusion volumetric analysis on estimated ischemic core and penumbral volumes

PLOS ONE

Dear Dr. Karhi,

Thank you for submitting your manuscript to PLOS ONE. After careful consideration, we feel that it has merit but does not fully meet PLOS ONE’s publication criteria as it currently stands. Therefore, we invite you to submit a revised version of the manuscript that addresses the points raised during the review process.

We look forward to receiving your revised manuscript.

Kind regards,

Ona Wu

Academic Editor

PLOS ONE

Reviewers' comments:

Reviewer's Responses to Questions

**Comments to the Author**

1. If the authors have adequately addressed your comments raised in a previous round of review and you feel that this manuscript is now acceptable for publication, you may indicate that here to bypass the “Comments to the Author” section, enter your conflict of interest statement in the “Confidential to Editor” section, and submit your "Accept" recommendation.

Reviewer #1: All comments have been addressed

Reviewer #2: (No Response)

2. Is the manuscript technically sound, and do the data support the conclusions?

Reviewer #1: Yes

Reviewer #2: Partly

3. Has the statistical analysis been performed appropriately and rigorously? 

Reviewer #1: Yes

Reviewer #2: No

4. Have the authors made all data underlying the findings in their manuscript fully available?

Reviewer #1: Yes

Reviewer #2: Yes

5. Is the manuscript presented in an intelligible fashion and written in standard English?

Reviewer #1: Yes

Reviewer #2: No

6. Review Comments to the Author

Reviewer #1: The authors responses to the comments and criticism were reasonable and the manuscript is updated appropriately. There is a minor error in the reference section. The items #27 and 36 refer to the same manuscript (Campbell BC, et al).

Reviewer #2: CTP thresholds are used to select stroke patients for EVT. In this revised paper the authors show wide disparities in infarct core and penumbral volumes when different software packages are used. The primary outcome is the accuracy of CTP thresholds with the final infarct volume measure by CT. There remain several important concerns that the authors’ need to address:

1) I would recommend the comparisons between the CTP thresholds be a second aim and not merely added in the supplement. The abstract, methods, results and discussion need to reflect this change.

2) The Bland Altman plots comparing CTP thresholds are incomplete. Typically, a BA plot shows the mean difference, 1.96 SD around that difference and the upper and lower margins. In the supplementary figure plots B and C are a duplicate.

3) When measuring the accuracy of CTP thresholds for core and penumbra with FIV typically a sensitivity analysis is performed not Pearson correlation.

7. PLOS authors have the option to publish the peer review history of their article (what does this mean?). If published, this will include your full peer review and any attached files.

Reviewer #1: No

Reviewer #2: No

---

## [Author Response · Author response to Decision Letter 1]

9 Mar 2021

Please see the separate document "Response to reviewers" for the full response letter.

Response to Reviewer #1:

Comment: The authors responses to the comments and criticism were reasonable and the manuscript is updated appropriately. There is a minor error in the reference section. The items #27 and 36 refer to the same manuscript (Campbell BC, et al). Response: The authors are thankful for the excellent feedback and comments giving us new viewpoints to our study and manuscript and also for future research on the subject. The error in the reference section has now been fixed. 

Response to Reviewer #2: 

CTP thresholds are used to select stroke patients for EVT. In this revised paper the authors show wide disparities in infarct core and penumbral volumes when different software packages are used. The primary outcome is the accuracy of CTP thresholds with the final infarct volume measure by CT. There remain several important 

concerns that the authors’ need to address: 

(The authors wish to point out that the study compares the results produced by a single software package using three different threshold levels.)

Comment: “1) I would recommend the comparisons between the CTP thresholds be a second aim and not merely added in the supplement. The abstract, methods, results and discussion need to reflect this change.” Response: Thank you for the valuable comments. To address the Reviewer’s concern, the comparison between the three CTP threshold settings has now been mentioned as the second aim of the study in Introduction, and furthermore, the Abstract and Discussion have been modified to better reflect the importance of the differences between the CTP thresholds. The suggested supplementary Bland Altman Plots have now been transferred from the supplementary materials to be part of the main manuscript in the Results section.

Comment: “2) The Bland Altman plots comparing CTP thresholds are incomplete. Typically, a BA plot shows the mean difference, 1.96 SD around that difference and the upper and lower margins. In the supplementary figure plots B and C are a duplicate.” Response: Based on the feedback, one of the two Bland Altman plots visualizing the setting pair S1 vs S2 when comparing Ischemic Core Volumes has now been excluded due to the similarities between the original panels B and C. 

As requested, the Bland Altman Plot figure now includes the numeral explanation for the lines for mean difference, and the 1.96 SD around the difference. Upper and lower margins for mean difference are now visualized with additional thinner dotted lines. In addition, an alternative and more detailed figure 2 (visualizing also upper and lower 95% confidence limits for the upper and lower “agreement limits”) is included based on a consultation of our biostatistician (see uploaded file titled: Alternative Fig 2). We are willing to use either of these alternative plots according to the Reviewer’s and Editor’s preferences. 

Comment: “3) When measuring the accuracy of CTP thresholds for core and penumbra with FIV typically a sensitivity analysis is performed not Pearson correlation.”

Thank you for the comment. The use of sensitivity analysis was considered by the authors when choosing between different methods for statistical analysis. Among these methods, Pearson Correlation was selected and has also been used in earlier literature, for example: Austein F, Riedel C, Kerby T, et al. Comparison of perfusion CT software to predict the final infarct volume after thrombectomy. Stroke. 2016; 47:2311–2317.

---

## [Editor Report · Decision Letter 2]

25 Mar 2021

Effect of different thresholds for CT perfusion volumetric analysis on estimated ischemic core and penumbral volumes

PONE-D-20-23892R2

Dear Dr. Karhi,

We’re pleased to inform you that your manuscript has been judged scientifically suitable for publication and will be formally accepted for publication once it meets all outstanding technical requirements.

Kind regards,

Ona Wu

Academic Editor

PLOS ONE
---

## [Editor Report · Acceptance letter]

6 Apr 2021

PONE-D-20-23892R2 

Effect of different thresholds for CT perfusion volumetric analysis on estimated ischemic core and penumbral volumes 

Dear Dr. Karhi:

I'm pleased to inform you that your manuscript has been deemed suitable for publication in PLOS ONE. Congratulations! Your manuscript is now with our production department. 

Kind regards, 

on behalf of

Dr. Ona Wu 

Academic Editor

PLOS ONE